# Potato (*Solanum tuberosum* L.) tuber-root modeling method based on physical properties

**Ping Zhao**[1], **Yue Tian**[1], **Yongkui Li**[1], **Guofa Xu**[1], **Subo Tian**[1]*, **Zichen Huang**[2]*

**1** College of Engineering, Shenyang Agricultural University, Shenyang, China, **2** Graduate School of Agriculture, Kyoto University, Kyoto, Japan

* tiansubo@syau.edu.cn (ST); huang.zichen.22c@kyoto-u.jp (ZH)

**Data Availability Statement:** All relevant data are within the manuscript and its Supporting Information files.

**Funding:** The present work is financially supported the National Natural Science Foundationof China

## Abstract

The development of tuber-root models based on the physical properties of the root system of a plant is a prominent but complicated task. In this paper, a method for the construction of a 3D model of a potato tuber-root system is proposed, based on determining the characterization parameters of the potato tuber-root model. Three early maturing potato varieties, widely planted in Northeast China, were selected as the research objects. Their topological and geometric structures were analyzed to determine the model parameters. By actually digging potatoes in the field, field data measurement and statistical analysis of the parameters were performed, and a model parameter database was established. Based on the measured data, the root trajectory points were obtained by simulating the growth of the root tips. Then MATLAB was used to develop a system that would complete the construction of the potato tuber-root 3D visualization model. Finally, the accuracy of the model was verified experimentally. Case studies for the three different types indicated an acceptable performance of the proposed model, with a relative root mean square error of 6.81% and 15.32%, for the minimum and maximum values, respectively. The research results can be used to explore the interaction between the soil-tuber-root aggregates and the digging components, and provide a reference for the construction of root models of other tuber crops.

## Introduction

The root system supplies water and minerals for a plant, in addition to anchoring the plant in the soil [1]. A visualized root system model is particularly important for mechanical component system dynamics that require modern agricultural machinery with high precision, high quality, strong adaptability, and the development of computer-aided design [2, 3]. Therefore, the development of on modeling methods based on the physical properties of the root system of a plant has become a popular though complicated topic in virtual plant research. The key to a model's success is closely related to the selection and determination of the model parameters. An effective method to determine model parameters is through measured data [4].

Potato (*Solanum tuberosum* L.) is the fourth largest crop in terms of production, and the most important non-grain food crop in the world [5–7]. The total potato production

(NSFC) of Zhao Ping, and the grant mumber is 51505305. Zhao Ping had an important role in the study design, data collection and analysis, decision to publish, and preparation of the manuscript.

**Competing interests:** The authors have declared that no competing interests exist.

worldwide is approximately 400 million tons per year; the most important potato producers are China, India, Russia, and the United States [8]. In this production chain, the stage of harvesting requires the greatest labor intensity. Damage to tubers during harvesting is one of the main causes of reduced potato quality and value [9], as well as tuber diseases during storage [10, 11]. Hence, the potato harvester has an important role in reducing labor intensity and tuber diseases during storage, and ensuring potato quality and value. The design of the digging components of the potato harvester directly influences the performance of the entire machine, and the interaction between the soil-tuber-root aggregates and the digging components is the theoretical basis for the design of the digging components. Therefore, it is extremely important to research and understand this interaction to establish a three-dimensional(3D) model construction of the potato tuber root system. Moreover, understanding the development of the potato root system has the potential to increase yield, optimize agricultural land use [7], and promote the genetic improvement of crops [12]. It also provides a reference for the stem-root model construction of other block, root, and bulb crops.

To date, there have been several studies on the root system model, including those for rice (*Oryza sativa*) [13–15], wheat (*Triticum*) [16–18], maize (*Zea mays*) [19–21], soybean (*Glycine max*) [22, 23], and other crops and plants[24]. These studies have focused on the tap or fibrous root system, which is composed of roots with a relatively simple configuration. Other researches regarding tuber-root models has simply focused on the yield and ignored the physical structure of tuber-root plants, such as cassava (*Manihot esculenta*) [24] and potato [25]. Although reference [26] established a tuber-root model regarding potatoes based on physical structure, the measured data of the physical structure of tuber-root were not from the same plant at different stage. Research [27] regarding the physical structure of the linear shape of yams (*Dioscorea spp*.) is difficult to use for describing the potato. Moreover, the true architecture of a root system, especially the rooting depth, has frequently been neglected [28].

In this research, three typical early-maturing potato varieties in Northeast China are selected as the research object, and a tuber-root modeling method based on the physical properties gathered during the potato-harvesting period is studied. The characterization parameters are determined according to the biological characteristics of the potato tuber-root system, measured in the field during the harvest period, and the root locus points are obtained by simulating the growth of the root tip. A visual system of the potato tuber-root system model is established using MATLAB. Because the structure description and parameter measurement of the root system are based on the natural growth state of the soil, and the parameter description is based on the statistical theory, the description of the objective growth law is more scientific. The topology is based on the standard algorithm design of data structure, which is more conducive to later algorithm development.

## Materials and methods

### Materials

The research object of the virtual crop was the potato tuber-root system. Fujin, Zaodabai, and Helanshiwu are early-maturing potato varieties widely grown in Northeast China and were selected as examples in this research. These potatoes were cultivated in late April and harvested in July. The field data measurement time was the potato harvesting period. The test tools for tuber-root characterization parameter testing included a shovel, brush, ruler (1 mm accuracy), vernier caliper (0.01 mm accuracy), protractor (1°accuracy), camera (Canon EOS 70D), and computer. The software for the data processing was SPSS (Statistical Product and Service Solutions, Version 22.0), and for visual modeling was MATLAB (2018b, student version).

## Methods

To begin with, based on an analysis of the growth characteristics of the potato tuber-root system, the model characterization parameters were analyzed and determined from two aspects: the topological and geometric structure. The model parameter database was developed using the measuring, counting, analyzing, and digging methods in the field. Then, according to the growth structure characteristics of the tuber-root system, which can be divided into three categories: seed-root, seed-tuber-underground stem-stolon-tuber, and underground stem-stolon root, the data structure and storage structure were established. The single root was formed based on the measured data, and the locus point of the root was obtained by stimulating the growth of the root tip. The model building system was designed in order to complete the construction of the potato tuber root 3D visualization model in MATLAB. Finally, the root depths of the three kinds of object simulation and measurement values were compared and analyzed. The accuracy and effectiveness of the model were verified by relative root mean square error (RRMSE) and the scatter diagram of the relationship between the simulation and measurement values.

## Determination and acquisition of characterization parameters of potato tuber-root model

### Structure of potato tuber-root model

The potato tuber-root system is developed from a seed potato. The seed potato grows downward in its seminal roots which absorb and deliver nutrition, and repair the plant. The seed potato grows upward in its underground stem (main stem) that breaks through the soil to form the plant. Creeping roots and stoloniferous stems originate from underground stems. Two sided lateral roots (fibrous roots) grow on the creeping and seed roots. The end of the creeping stem expands to form a tuber. The potato tuber-root system (Fig 1) consists of an underground stem, creeping stem, seed potato, seminal root, fibrous root, tuber and creeping roots. When a potato tuber-root model is established, we ignore the fibrous roots because they are overly slim and unlikely to influence the interaction force between the digging part of the potato and soil-tuber-root system.

### Determining characterization parameters of potato tuber-root model

The characterization parameters of a model provide the basis for digitizing the potato rhizome entities into a computer model. They have a decisive role in the root morphology and are important parameters necessary to ensure the accuracy of the model. This paper determined the characterization parameters of the model from two aspects, the topology and geometric structures.

**Potato tuber-root topology.** Topological structure considers only the positional relationship among objects, and not their shape and size. To achieve the modeling requirements, the geometric center of the seed potato is used as the coordinate origin to establish a Cartesian 3D coordinate system XY plane. This allows the simplification of the potato tuber-root topology (Fig 2).

It can be observed from Fig 2, that the characterization parameters of the potato root topology are the axial angle $\theta$ (angle between the root or stem $i$, and underground stem), radial angle $\eta$ (angle between the projection of the root or creeping stem $i$ on the $XY$ plane and $Y$-axis), distance $h$ (between the seed potato and rooting point of the root or creeping stem $i$), and the number of tubers. Because the root system is curved under natural conditions, the

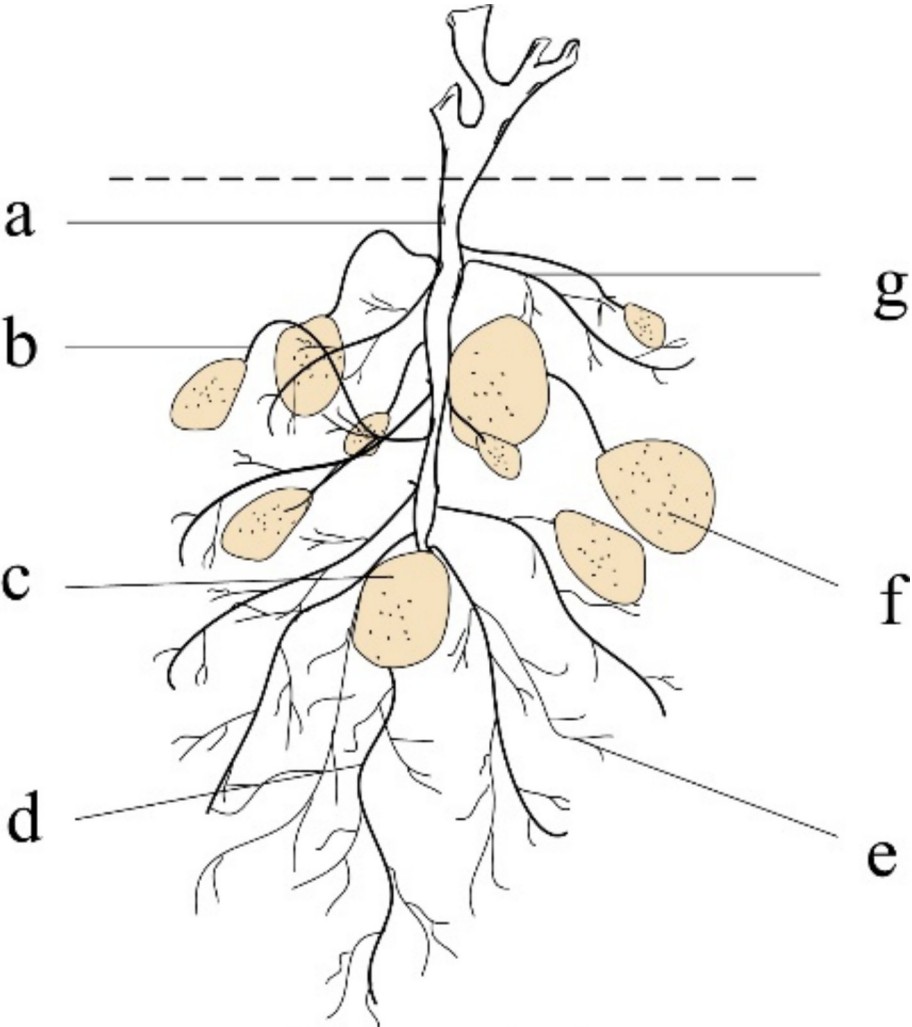

**Fig 1. Potato root organization.** a: Underground stem; b: Creeping stem; c: Seed potato; d: Seminal root; e: Fibrous root; f: Tuber; g: Creeping root.

axial and radial angles of the root system are taken from the root as the initial angles during the measurement.

**Geometric structure of potato tuber-root system.** The tuber-root system of potatoes can be divided into roots, main stem, and tubers according to the geometric structure. The roots include seminal roots, creeping roots, and creeping stems. As indicated in the geometric structure (Fig 3), the characterization parameters of the root geometric structure are initial radius $r_{g0}$, root tip radius $r_{g1}$, soil depth $d$, root length $l$, and total deflection angle $\Phi$.

The main stem refers to the underground stalk of the potato plant (Fig 4). The main characterization parameters are bottom radius $r_{j0}$, top radius $r_{j1}$, and height $h_j$. Tubers include mainly seed potatoes and tubers. Tubers include mainly seed potatoes and tubers. To reduce the simulation complexity, this study assumes that the surface of the tubers is uniform, and that its structural shape can be divided into spherical, ellipsoid and elongated. The geometric parameters of the tubers are length $L$, width $W$, and Highly $H$.

In summary, the characterization parameters of the potato tuber-root system model are summarized in Table 1.

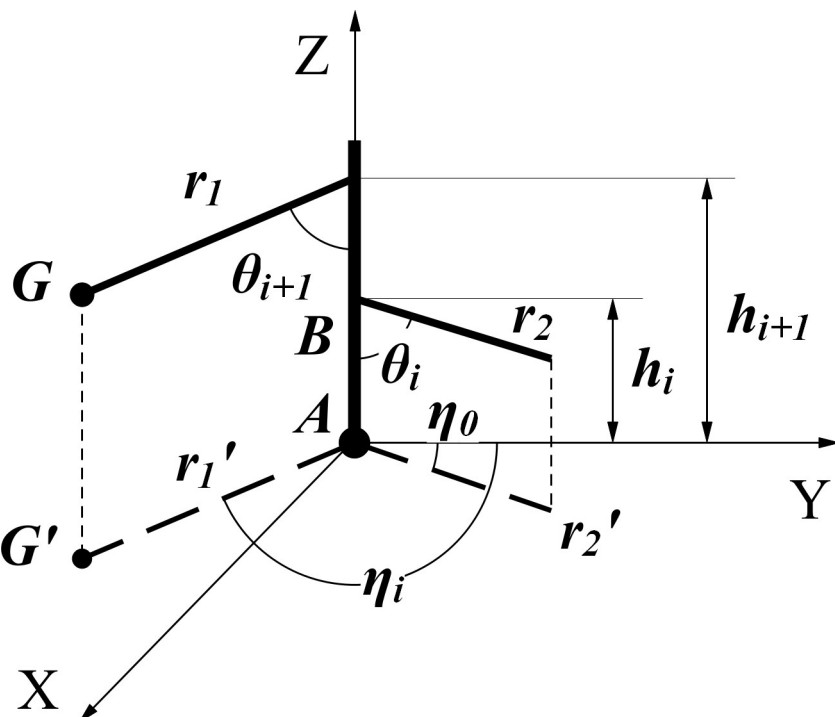

**Fig 2. Schematic diagram of potato tuber-root topology.** *A*: Seed potato; *B*: Underground stem; *G*: Tuber; *G'*: Tuber projection; $r_1$: Root 1; $r_2$: Root 2; $r_1$': Root 1 projection; $r_2$': Root 2 projection.

## Measurement and analysis of characterization parameters of potato tuber-root system model

**Measuring parameters of model characterization.** The experiment was conducted at the potato-planting base in Jianping County, Chaoyang City, Liaoning Province, PR China. This is the main potato production area in Liaoning Province. The test field area was 2000 m². The test objects were the representative early maturing varieties Fujin, Zaodabai and the Helanshiwu.

During the potato-harvesting period, a field experiment was undertaken to measure the characterization parameters. To ensure that the test results were more coordinated with the

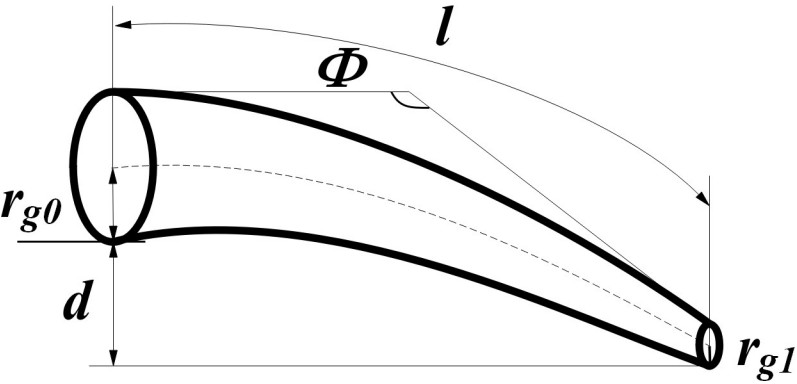

**Fig 3. Root geometry diagram.**

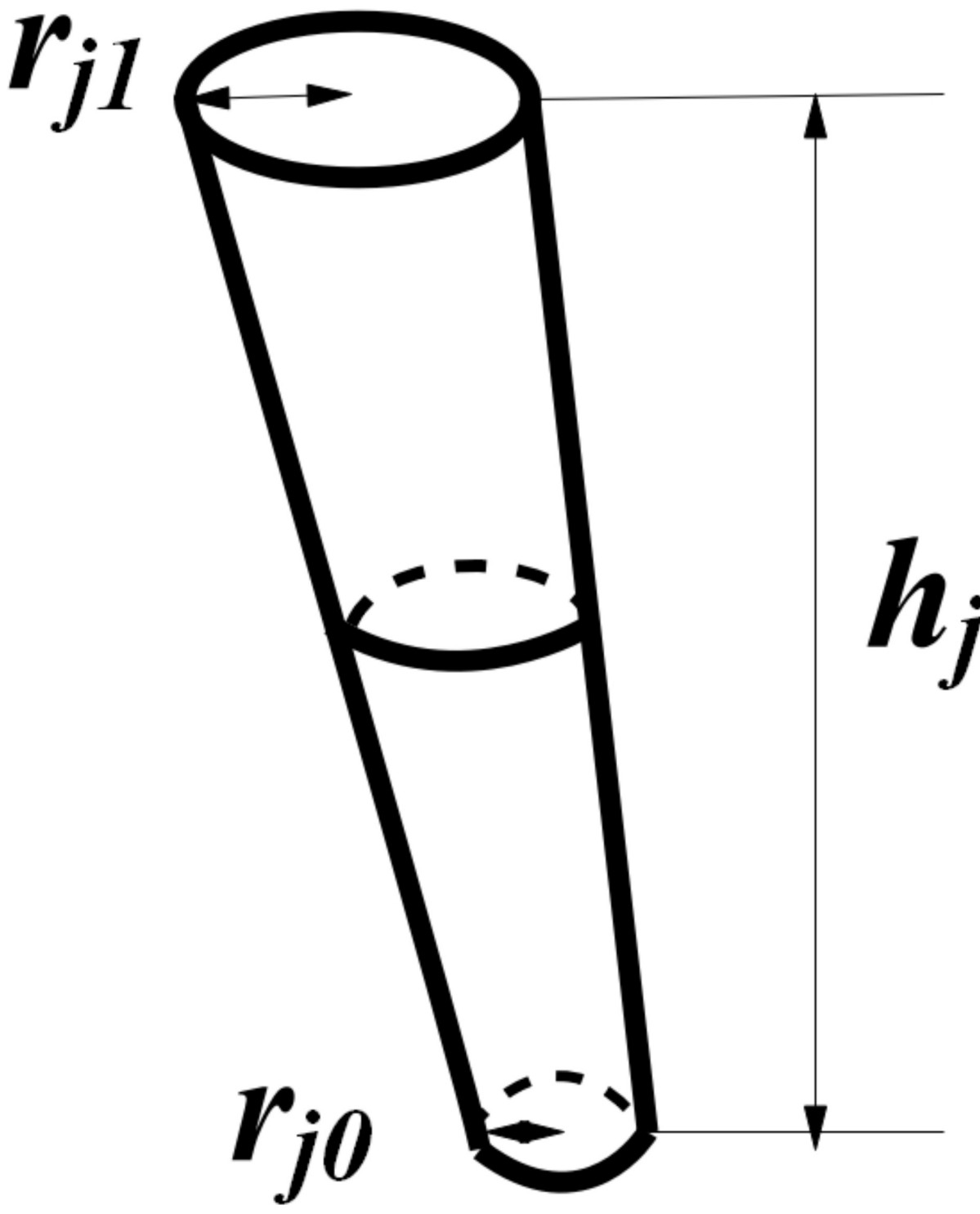

**Fig 4. Schematic diagram of the main stem geometry.**

actual situation, based on statistical theory, each plant was randomly selected from 50 plants with an acceptable growth condition, as a measurement sample. In this experiment, the excavation method was used. First, the stems and leaves above the ground were excised. Based on

**Table 1. Characterization parameters of potato tuber-root model.**

| Model category | Roots | Tubers | Main stems |
|---|---|---|---|
| Characterization parameters | Axial angle ($\theta$) | Length ($L$) | Bottom radius ($r_{j0}$) |
| | Radial angle ($\eta$) | Width($W$) | Top radius ($r_{j1}$) |
| | Initial radius ($r_{g0}$) | Height ($H$) | Height ($h_j$) |
| | Root tip radius($r_{g1}$) | Number ($N_k$) | - |
| | Root length ($l$) | - | - |
| | Root depth ($d$) | - | - |
| | Total deflection angle ($\Phi$) | - | - |
| | Number ($N_g$) | - | - |
| | Distance to seed potatoes ($h_g$) | - | - |

the exploration experience, trenches were made at a distance of 0.3 m from the plant (ditch width 0.1 m, depth 0.9 m, length 0.3 m). Then, gentle digging was initiated downward from the topsoil. When approaching the uppermost root system, the soil was brushed in a direction parallel to the rhizome until the soil profile of the root system was exposed (Fig 5). The roots were labeled and the original parameters measured, including the axial angle (the actual measurement was the supplementary angle of the axial angle), depth of the soil, length, axial deflection angle $\Phi$, radial deflection angle, root radius, and tip radius (used to calculate the radius change coefficient). Then, grooves of the same size were dug on the other three sides. After removing and cleaning the entire root system (Fig 6) with a brush, the parameters of the tubers and roots were measured, including the number of roots, stems, and tubers, the distance from each root and stem node to the seed potatoes, and the geometric dimensions of the underground stems and tubers. Each angle was measured with a steel ruler and protractor. The depth of the soil was measured with two steel rulers. The length was measured with a soft ruler. The geometric size of the underground stem was measured with a steel ruler and a digital Vernier caliper. The tuber size was measured by a machine vision system [29].

**Data processing and analysis.** The statistical analysis was analyzed using SPSS software to determine the distribution model or distribution range of each parameter by descriptive statistics, comparing means, correlation analysis, regression analysis, and non-parametric tests.

The pre-analysis indicated that certain characterization parameters of potato tuber-root have significant distribution characteristics, and others have a non-significant distribution. To facilitate the use of these parameters in modeling, the normal random function (NORMRND) was used to express the parameters with significant normal distribution characteristics. For parameters without significant distribution characteristics, the probability distribution function was used to express the parameters. For example, the probability distributions of the initial axial angle of the creeping Zaodabai root were calculated as follows. The corresponding distributed probability P of the initial axial angle APz in the range of [70°, 85°], [86°, 100°], [101°, 115°], [116°, 130°], [131°, 145°], and [146°, 160°] were 0.12, 0.36, 0.14, 0.16, 0.16, and 0.06, respectively. To calculate the initial axial angle APz, a random value between [0, 1], P = rand (1) was generated; if P was in [sum (0: Pi-1), sum (0: Pi)], the initial axial angle APz = unifrnd (ai, bi) (i = 1, 2,..., n and P0 = 0).

According to the above method, all the parameters of the root systems of the three potato varieties were organized and summarized, and all characterization parameter databases were established for modeling. There were sufficient root class samples because each plant included more than one root. Therefore, 50 samples of each root type (seed root, creeping root, creeping stem) were used for the model verification experiments.

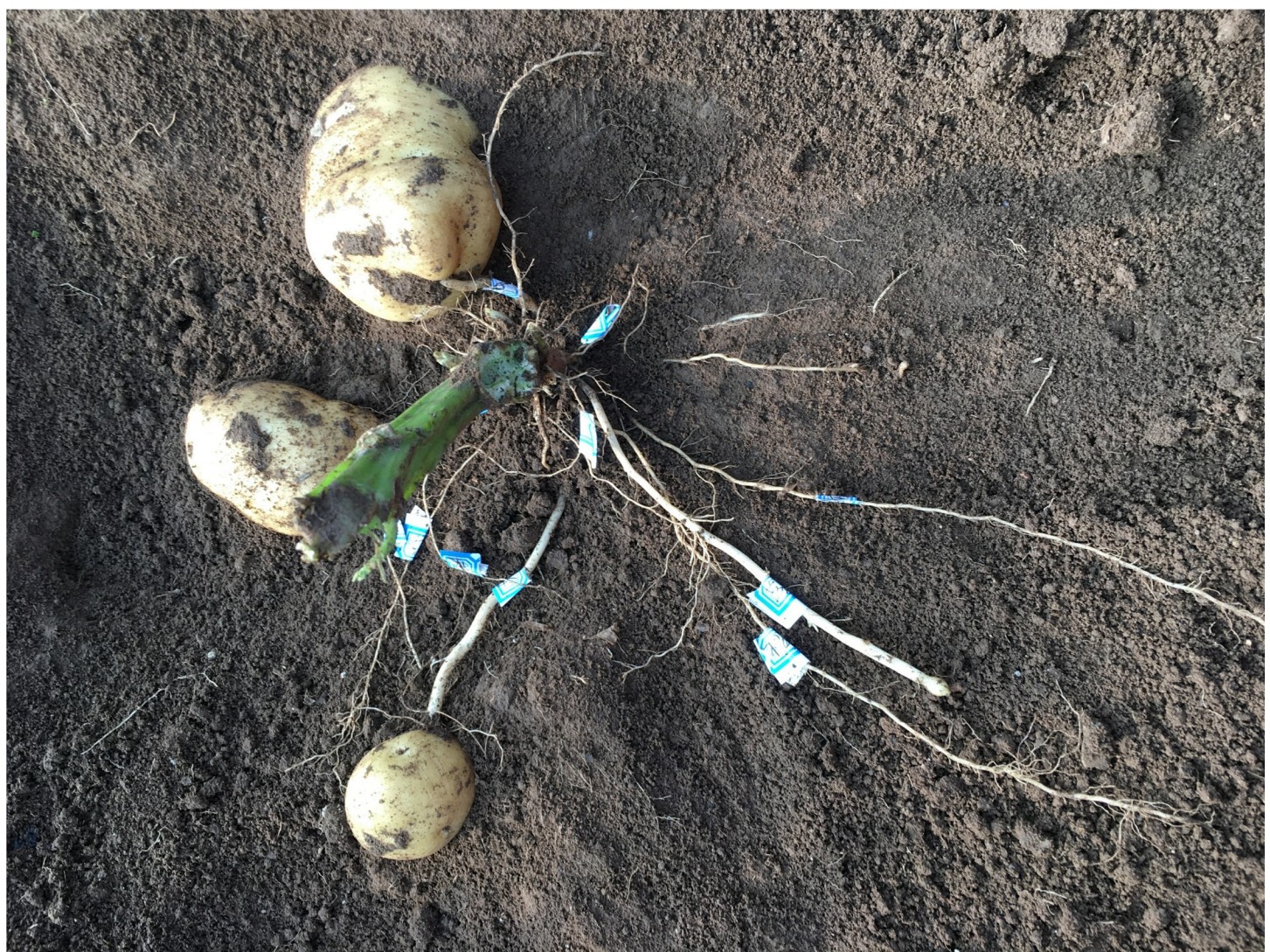

**Fig 5. Open root system.**

## 3D modeling of potato tuber-root system

### Data structure and algorithm

**Data and storage structure.** To begin, the root and tube types of a single potato tuber-root system model is defined as follows:

Seed potato: A, A = {a};

Underground stem: B, B = {b};

Seminal root: C, C = {c1, c2, c3, . . ., cn};

Creeping stem: D, D = {d1, d2, d3, . . ., dn}; and

Creeping root: E, E = {e1, e2, e3, . . ., en};

Tuber: F, F = {f1, f2, f3, . . ., fn}.

To conform to the basic model of root growth and truly reflect the topological structure of different potato rhizomes, this study established the overall logical structure and

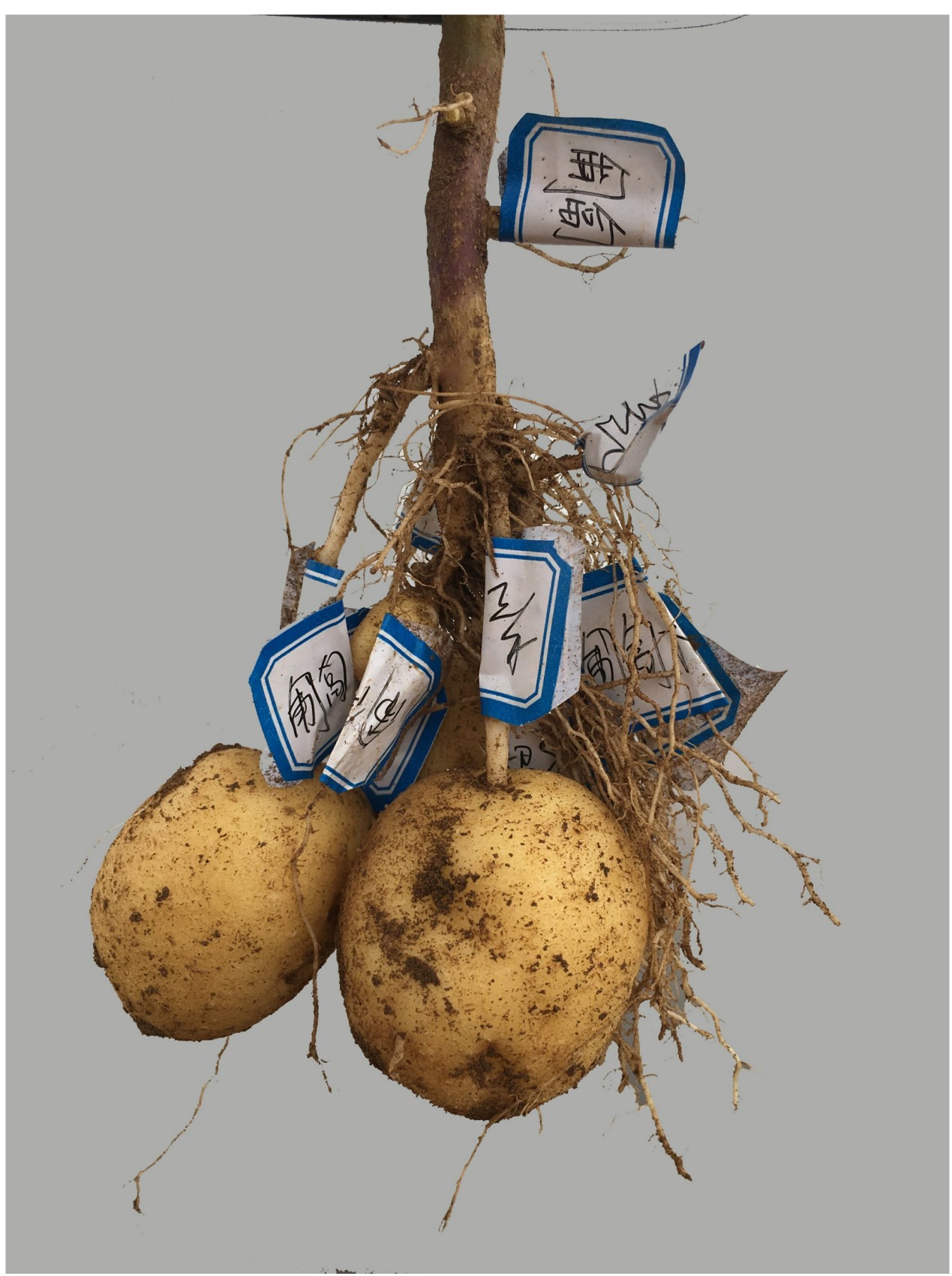

**Fig 6. Complete Zaodabai root system.**

corresponding storage structure (Fig 7) of the potato tuber-root model based on the rules of potato root growth.

According to the growth structure of the different roots and stems of the potatoes, the root-tuber system of potatoes can be divided into three parts, namely: (1) seed potato-seminal root; (2) seed potato-underground stem-creeping stem-tuber; and (3) underground stem-creeping root. The growth of the root system has a clear grading phenomenon. Hence, this paper used a tree structure to represent the relationship among the rhizomes. Since the seed potato-seminal root and underground stem-creeping root parts have the same structure when the morphology, geometric characteristics, and other factors are not considered, they can be modeled according to the secondary structure (Fig 8(A)). The seed tube-underground stem-creeping stem-tuber part was designed separately as a four-level structure (Fig 8(B)) for modeling.

Each node in Fig 8 represents a stem or a root. Using Fig 8(A) as an example, the seed potato used as the root node forms the first level, while the n seed roots M1, M2 . . . Mn form the second level. All nodes in the second level are sibling nodes, and are the child nodes of the potato (M represents the series; n represents the degree of the node). To facilitate the coverage of all root systems and avoid subsequent cumbersome procedures for obtaining root data, the storage structure adopted in this study is a child chain notation, as displayed in Fig 9(B).

A single root or stem can be regarded as a linear table structure connected by N growth units. Using a seed root as an example, its logical structure and corresponding storage structure are displayed in Fig 10.

**Algorithms for generating root and stem models.** According to the geometric characteristics of each component of the potato tuber-root system, the model is divided into four categories: root, seed potato, underground stem, and tuber.

1. Root generation algorithm. In the process of potato root growth, the spatial position coordinate of the seed potato is set as $(0, 0, z_0)$, which is the starting node of the underground stems and seminal roots, where $z_0$ is the buried depth of the seed potatoes. According to

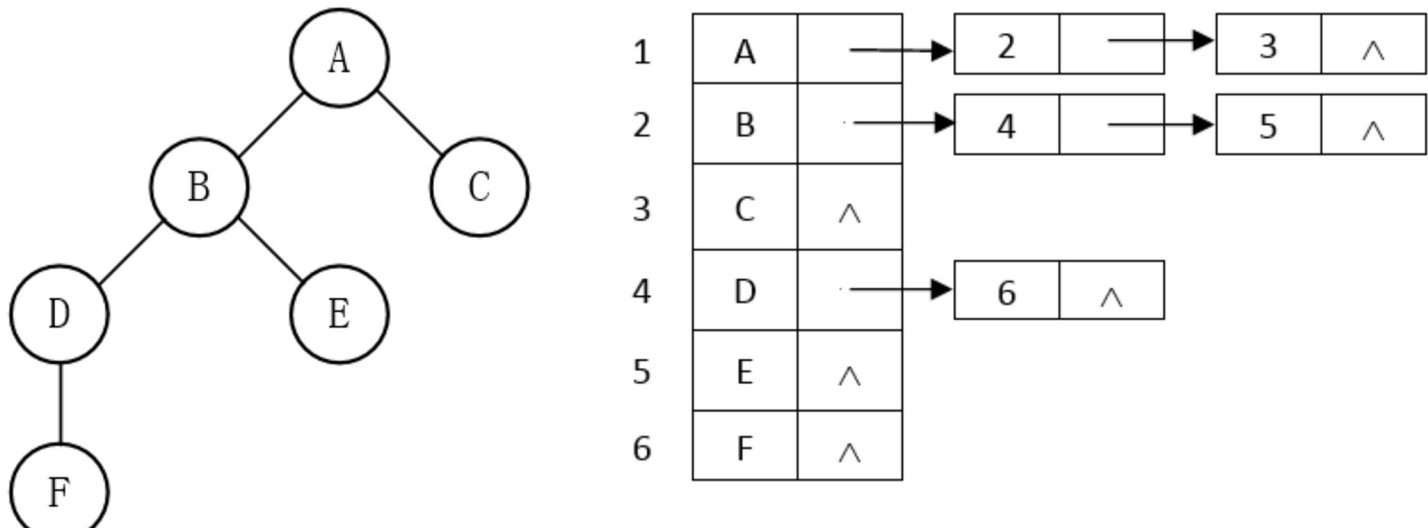

**Fig 7. Overall structure of potato tuber-root model.** (a) Logical structure; (b) Storage structure.

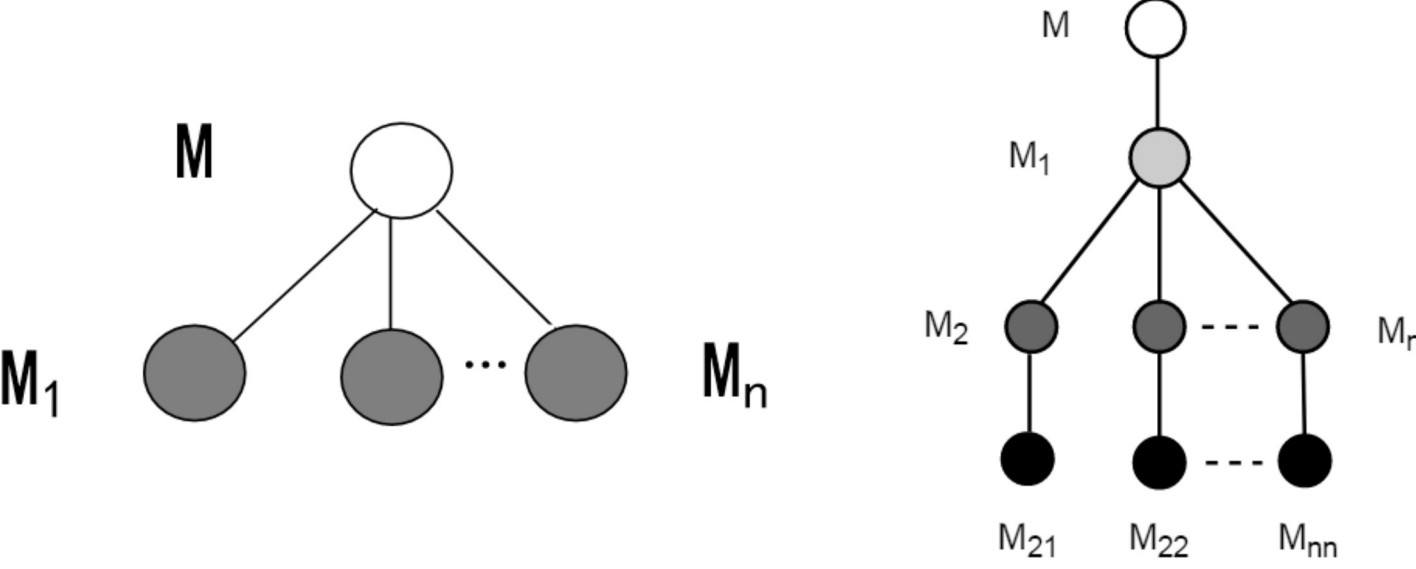

**Fig 8. Logical structure among rhizomes.** (a)Two level; (b) Four level.

the topological structure, creeping roots and creeping stems are derived from underground stems. Hence, the starting nodes of the creeping roots and creeping stems are on the root axis of the underground stems, and the specific location is determined by the measured distance from each root node to the seed tube. The starting node of the tuber is at the terminal node of the creeping stems.

Influenced by several factors, including geotropism, growth randomness, and soil resistance, the direction of the potato root growth can deflect at any time, making the growth trajectory curved, as indicated in Fig 11.

Suppose that the starting node of a root is A($x_1, y_1, z_1$), and the original growth trajectory is to grow a unit length $l$ along the $Y$-axis to the point B($x_1, y_1 + l, z_1$), as indicated in Fig 12. Considering the deflection under the influence of different factors and adding rotation $\gamma°$ around the Z axis and $\theta°$ around the X axis, the actual growth point C ($x_2, y_2, z_2$) can be obtained by the following equation,

$$[x_2, y_2, z_2, 1] = [x_1, y_1, z_1, 1] \bullet \begin{bmatrix} 1 & 0 & 0 & 0 \\ 0 & 1 & 0 & 0 \\ 0 & 0 & 1 & 0 \\ 0 & l & 0 & 1 \end{bmatrix} \bullet \begin{bmatrix} \cos\gamma & \sin\gamma & 0 & 0 \\ -\cos\theta \times \sin\gamma & \cos\theta \times \cos\gamma & \sin\theta & 0 \\ \sin\theta \times \sin\gamma & -\sin\theta \times \cos\gamma & \cos\theta & 0 \\ 0 & 0 & 0 & 1 \end{bmatrix} \quad (1)$$

In the actual simulation, the smaller the value of the root growth unit length $l$, the more realistic the model. The radial deflection angle $\gamma$ and axial deflection angle $\theta$ are selected from the established database.

From the above method of simulating the growth of the root tips, as long as the starting node of the root system and the termination conditions of growth can be determined, the growth trajectory of the root tip can be simulated. In this paper, the length of the root system was used as the termination condition, and variable data was used as the storage address to store the cyclically updated coordinates of the root tip. When the growth length was satisfied and the growth is stopped, the entire root axis was rotated by the initial axial angle and initial radial angle, and the final root axis coordinates were stored by setting storage unit $N$. Finally,

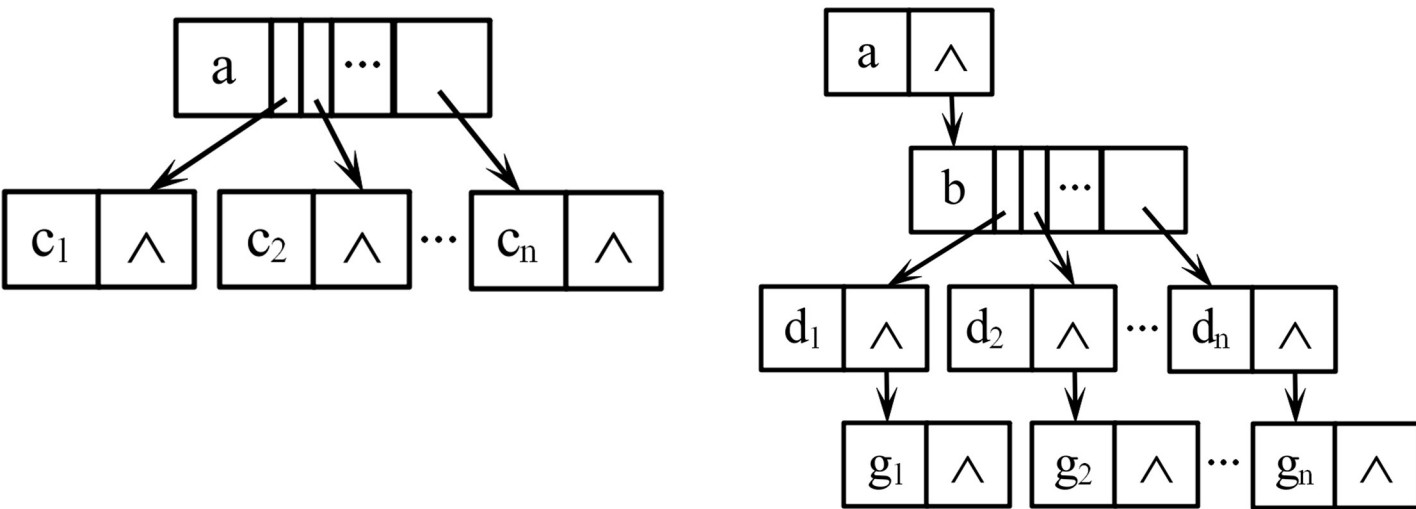

**Fig 9. Storage structure of relationships among rhizomes.** (a) Two level; (b) Four level.

the 3D root axis graph was drawn by plot3 (MATLAB function). The required parameters included the initial position $PP$, initial axial angle AP, initial radial angle $JP$, length $LP$, axial deflection angle $aP$ and radial deflection angle $jP$.

2. Seed potato generation algorithm. Seed potatoes are the seeds of the potatoes. Seed potatoes at maturity can be approximated as a semi-ellipsoid shape, achieved by the three parameters of length $LS$, width $WS$, and height $HS$ in the model construction.

3. Underground stem generation algorithm. Underground stems are shaped similar to a round table with a uniformly varying radius; however, in reality the surface of underground potato stems is rough and grows irregularly. To be closer to reality, this paper used the method of round table stitching to construct the underground stem model; that is, the round tables whose centers were not on a straight line were spliced together to simulate the effect of a rough surface. The required function here is the Cone function, and the required parameters include height $HD$, initial position (the origin by default), initial radius $RD$, and radius change rate $rD$.

4. Tuber generation algorithm. Tubers are divided into three shapes: elongated, ellipsoidal, and spherical, depending on the aspect ratio of the tubers. They can be represented by the

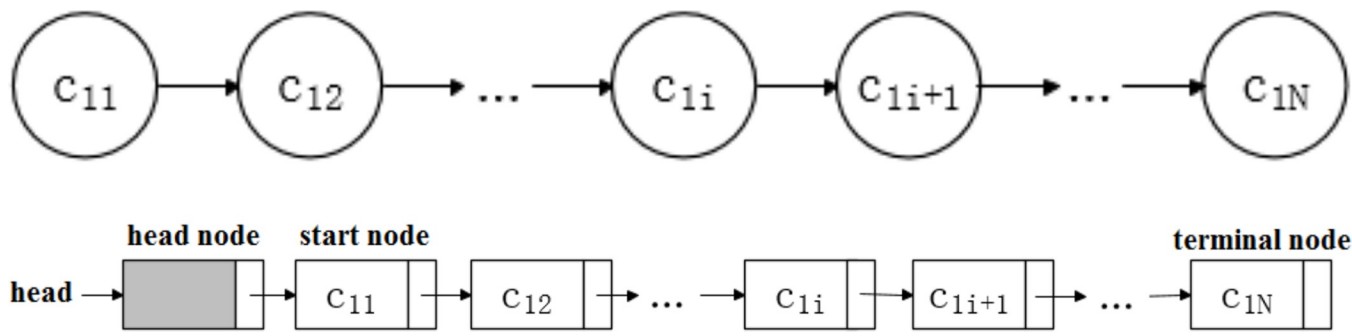

**Fig 10. Structure of single seed root.** (a) Logical structure; (b) Storage structure.

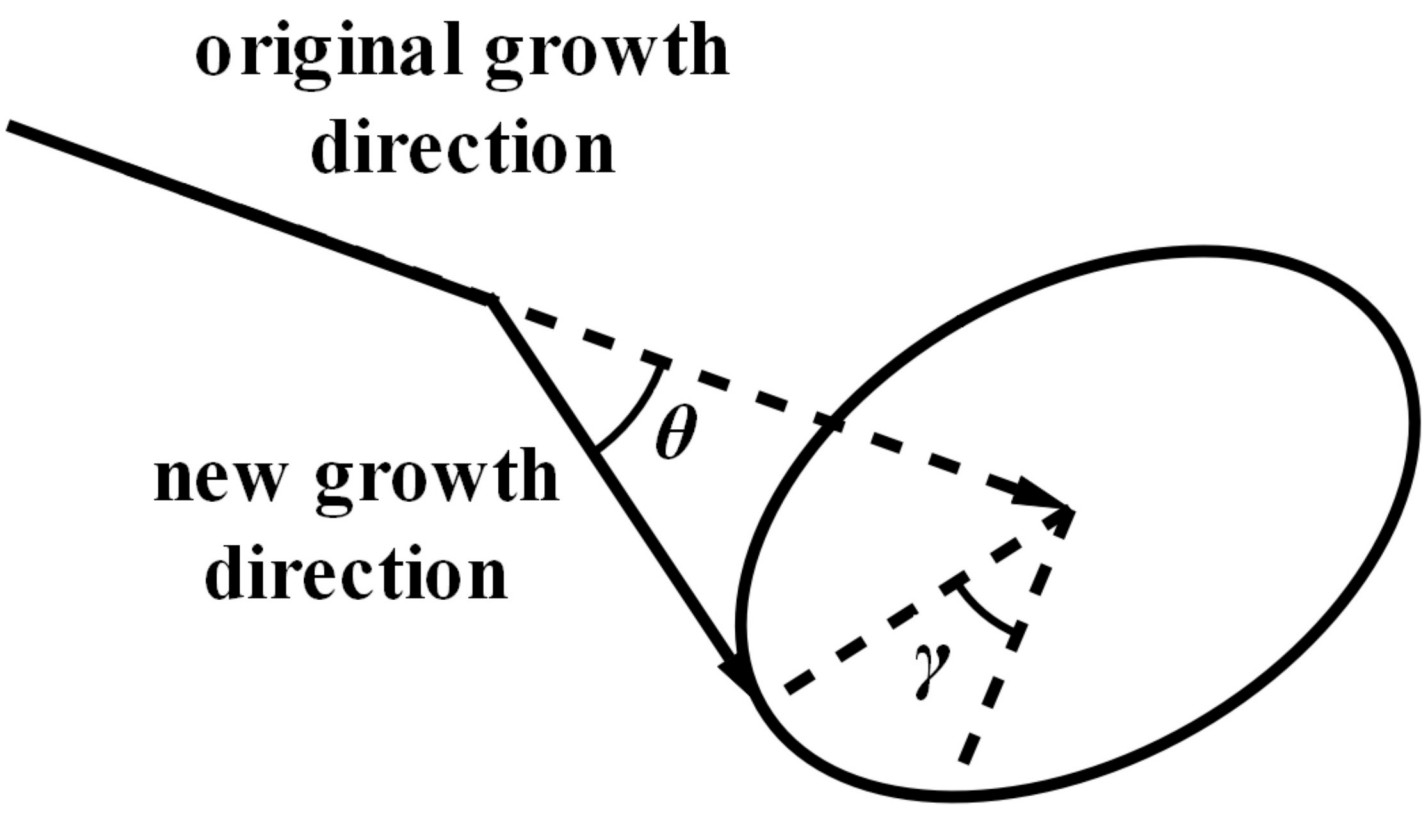

**Fig 11. Root growth direction.**

Ellipsoid function with the required parameters including the center point *PK*, length *LK*, width *WK* and height *HK*. Simulated tubers of different shapes are displayed in Fig 13.

## Program design of potato tuber-root system model construction

**System design.** According to object-oriented programming theory, this study used inheritance to organize different model types with the common attribute of the components of a complete model being the variety. Therefore, an abstract base class root system was established. Subclasses included seed potatoes, seminal roots, underground stems, creeping roots, creeping stems, and tubers. Different types of unified modeling language (UML) are displayed in Fig 14.

**Process design.** According to the data structure designed by the structure of the potato tuber-root model, the overall flow chart of the model construction is displayed in Fig 15. Each class of the model has its own modeling program, the principle is the same; however, the parameters are marginally different. Using creeping roots as an example, the construction process for all creeping roots in a potato root model is displayed in Fig 16. To improve the accuracy of the program identification, the potato varieties are replaced with numbers: 1-Zaodabai; 2-Helanshiwu; 3-Fujin.

## Case reconstruction and model verification

**Instance refactoring.** To facilitate the retrieval of the models, a graphical user interface (GUI) was designed using MATLAB. Users are only required to input the variety

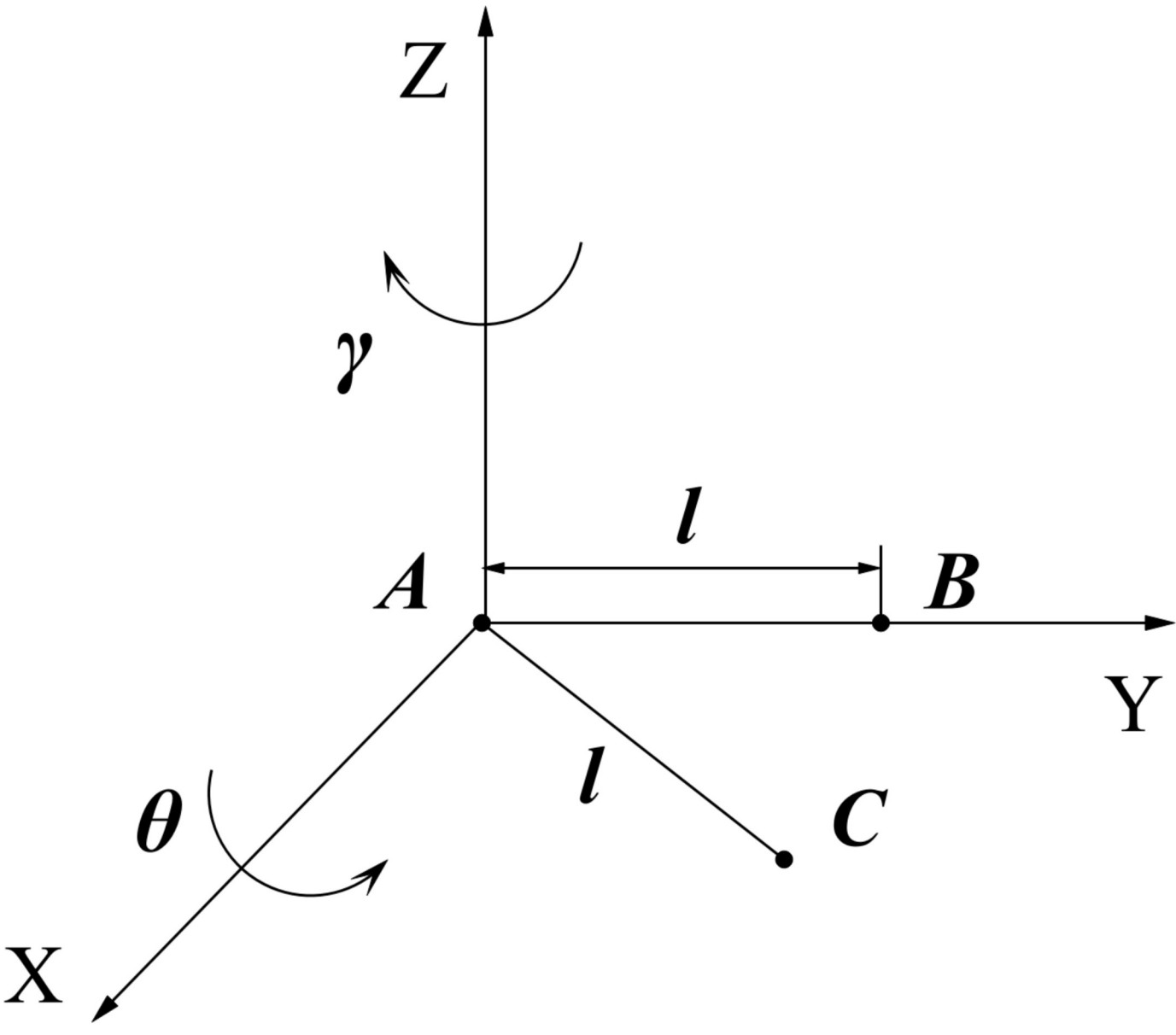

**Fig 12. Root growth trajectory.**

code to construct a 3D model of the potato tuber-root system. According to the method proposed in this study, examples of the three varieties of Zaodabai (Fig 17(A)), Helanshiwu (Fig 17(B)), and Fujin (Fig 17(C)) of the potato tuber-root model are displayed in Fig 17.

**Model validation.** In this paper, the root depth measured in "Determination and acquisition of characterization parameters of potato tuber-root model" was used to validate the consistency of the simulated values of the measured seminal roots, creeping roots, and creeping root depths of the three varieties with the measured values. Considering a relatively large distribution range of the root depth, RRMSE was used to test the accuracy of the model. The

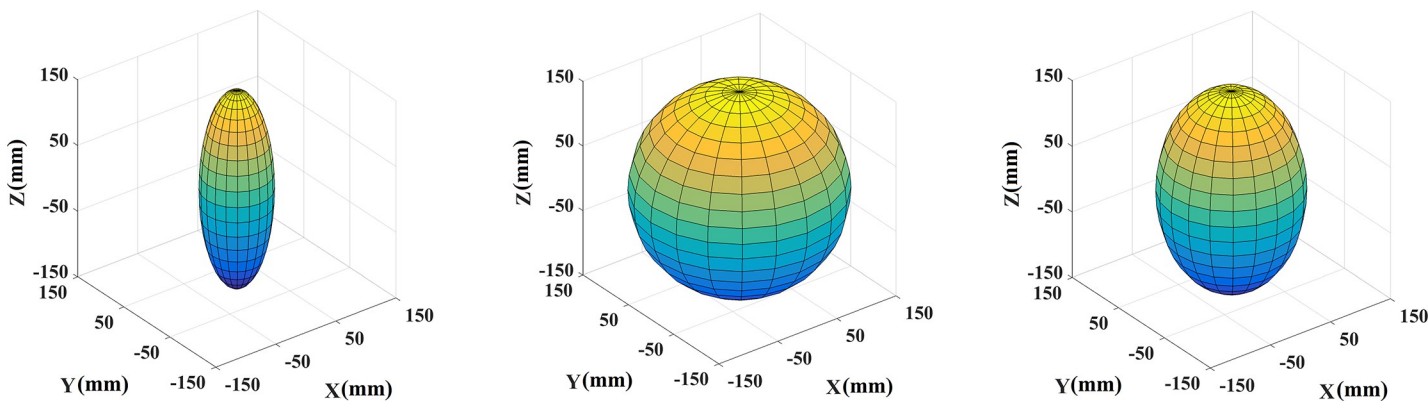

**Fig 13. Tubers of different shapes.** (a) Long potato; (b) Spherical potato; and (c) Ellipsoid potato.

calculation equation is.

$$RRMSE = \frac{\sqrt{\frac{1}{n}\sum_{i=1}^{n}(OBS_i - SIM_i)^2}}{\frac{1}{n}\sum_{i=1}^{n}OBS_i} \tag{2}$$

where $OBS_i$ is the measured value, $SIM_i$ is the simulation, and $n$ is a sample amount. The smaller the RRMSE value, the closer the measured value is to the simulation value. The evaluation standard of the RRMSE value for the simulation accuracy of the model can divide the

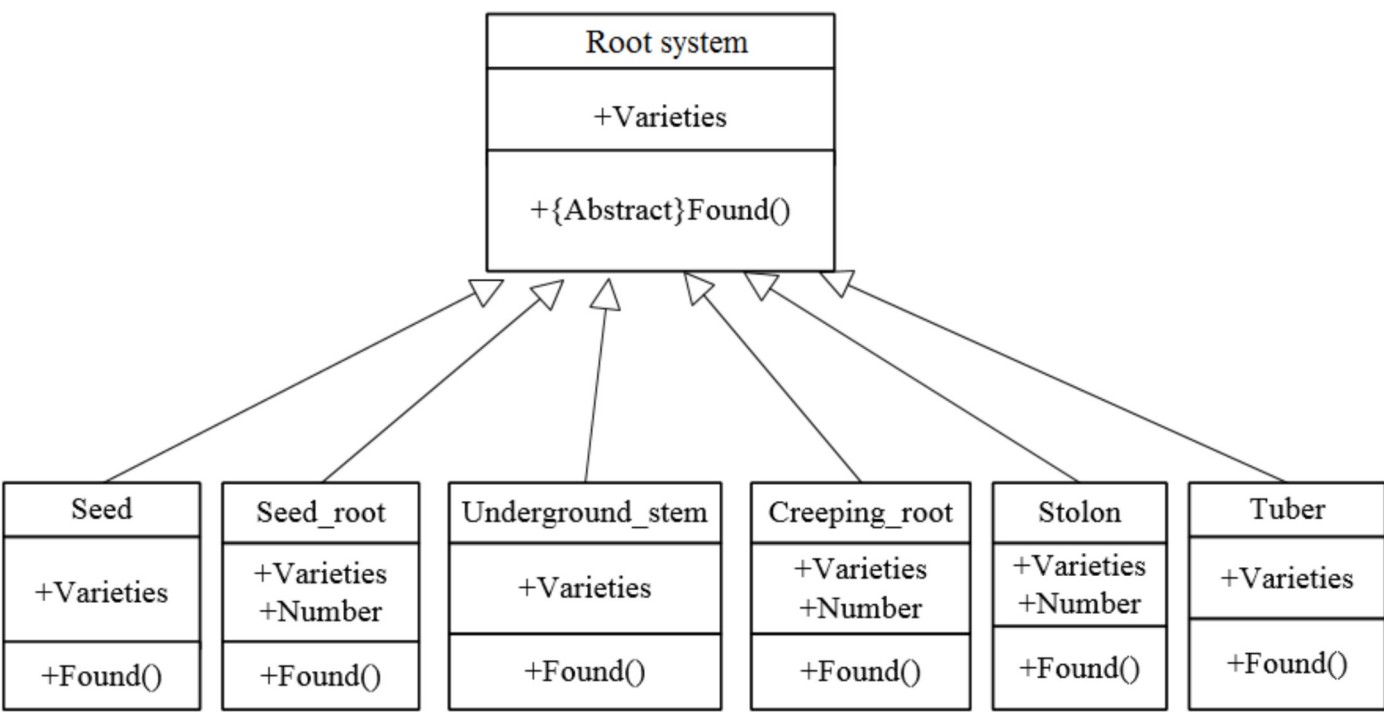

**Fig 14. Different types of organizational structure.**

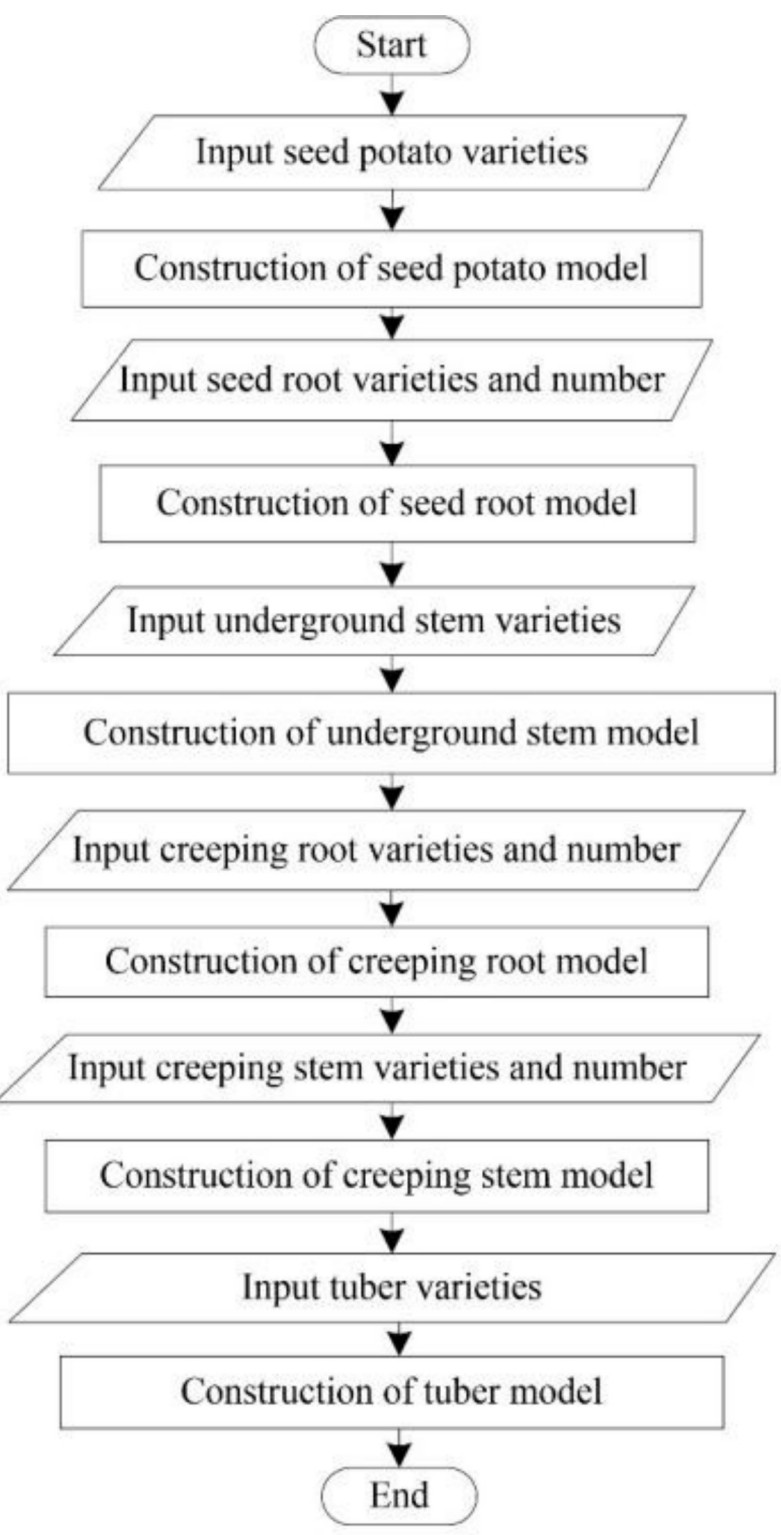

**Fig 15. Flowchart of building models.**

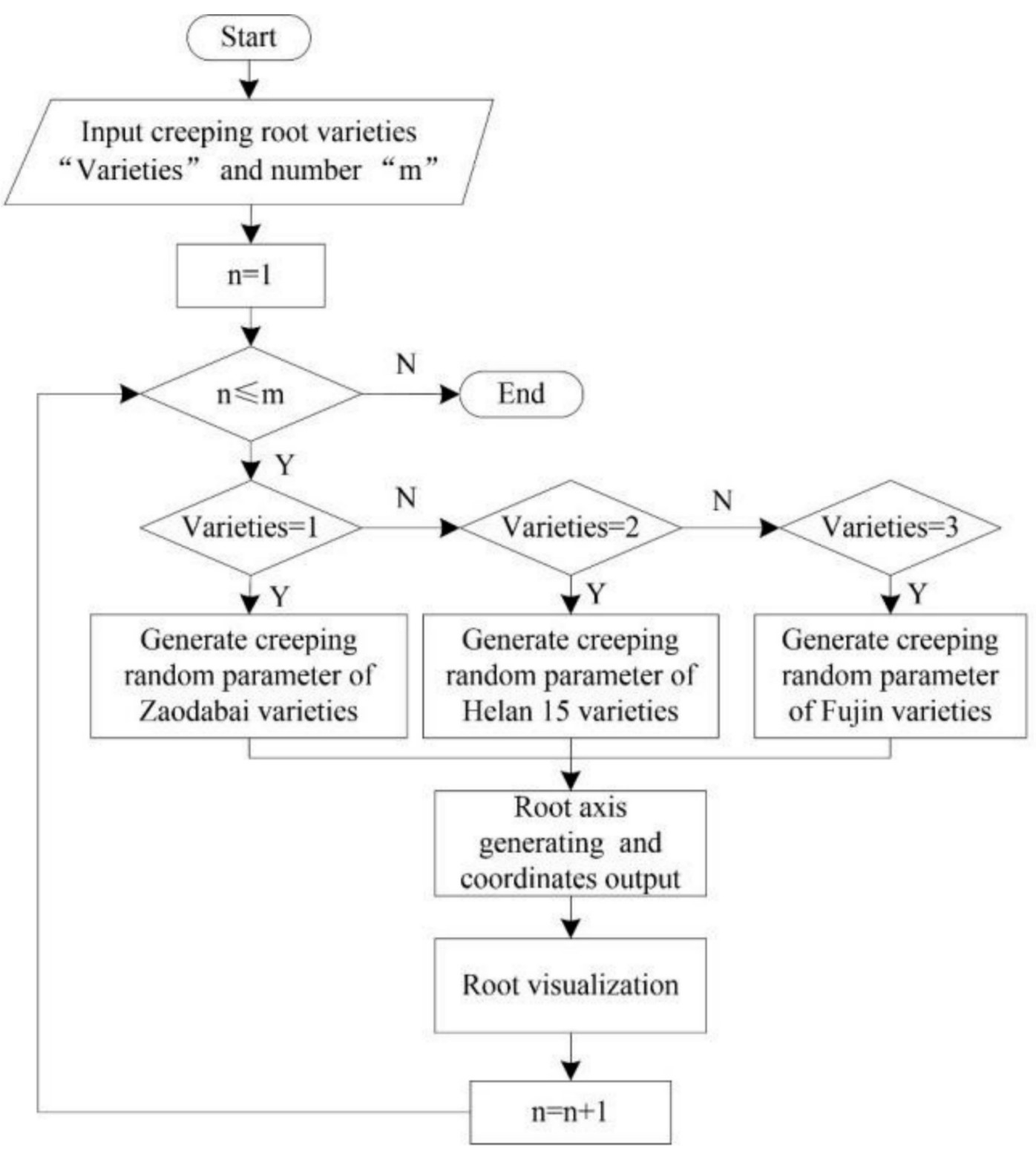

**Fig 16. Flow chart of multiple creeping root model construction.**

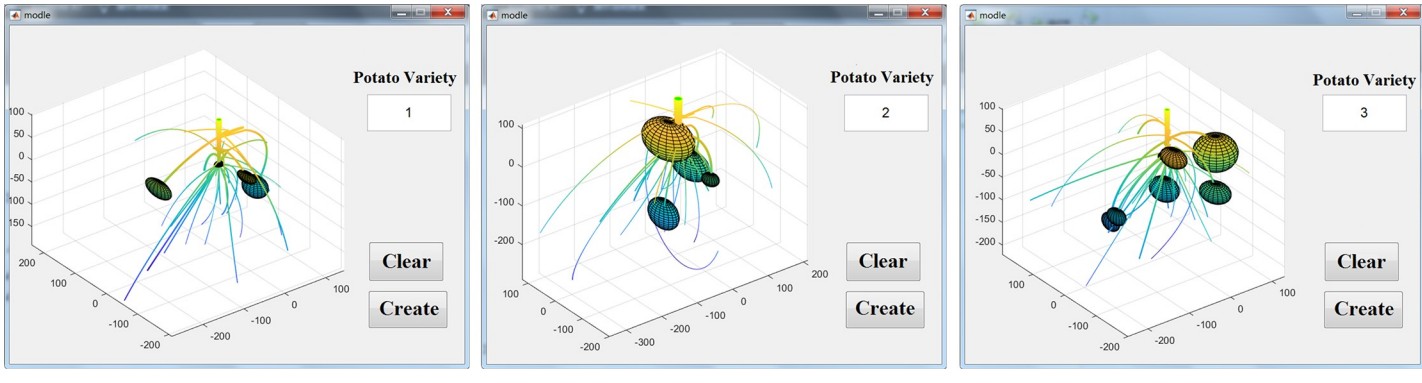

**Fig 17. Visualization of different potato models.** (a) Zaodabai; (b) Helanshiwu; and (c) Fujin.

simulation accuracy into four levels [30]. When RRMSE is less than 10%, it indicates that the consistency between the simulated and measured values is extremely significant. When RRMSE is between 10% and 20%, the consistency is significant. When RRMSE is between 20% and 30%, the simulation effect is general. When RRMSE is greater than 30%, it means that the deviation between the simulated and measured values is large, and the simulation effect is poor.

## Results and discussion

The actual measured value of the tuber-root models was determined using the reserved data of 50 samples. The simulated root depth of the tuber-root models was calculated by Eq (1). These are summarized in Table 2.

To verify the accuracy of the simulated model, a comparison of measured and simulated root depth was performed and is displayed in Figs 18–20.

An acceptable coherence between the measurements and simulations was found for these root depths. The Helanshiwu demonstrated superior global consistency compared to Fujin

**Table 2. Simulated and measured root depth.**

| Species | Sample No. | Seminal root | | Creeping root | | Creeping stem | |
|---|---|---|---|---|---|---|---|
| | | Measured value (mm) | Simulated value (mm) | Measured value (mm) | Simulated value (mm) | Measured value (mm) | Simulated value (mm) |
| Zaodabai | 1 | 94 | 96.87 | 35 | 33.42 | 40 | 36.36 |
| | 2 | 98 | 90.52 | 180 | 190.47 | 30 | 33.96 |
| | 3 | 133 | 127.64 | 96 | 93.21 | 68 | 60.45 |
| | . . . | . . . | . . . | . . . | . . . | . . . | . . . |
| | 50 | 80 | 86.38 | 140 | 144.18 | 60 | 50.36 |
| Helanshiwu | 1 | 142 | 124.87 | 210 | 188.67 | 30 | 27.85 |
| | 2 | 104 | 100.22 | 170 | 177.56 | 84 | 80.47 |
| | 3 | 86 | 78.47 | 80 | 85.49 | 38 | 48.39 |
| | . . . | . . . | . . . | . . . | . . . | . . . | . . . |
| | 50 | 130 | 110.94 | 210 | 209.46 | 48 | 45.26 |
| Fujin | 1 | 78 | 77.23 | 96 | 90.86 | 47 | 52.48 |
| | 2 | 124 | 136.85 | 140 | 132.23 | 80 | 90.27 |
| | 3 | 86 | 94.81 | 200 | 204.98 | 93 | 100.46 |
| | . . . | . . . | . . . | . . . | . . . | . . . | . . . |
| | 50 | 103 | 108.45 | 146 | 144.48 | 60 | 58.46 |

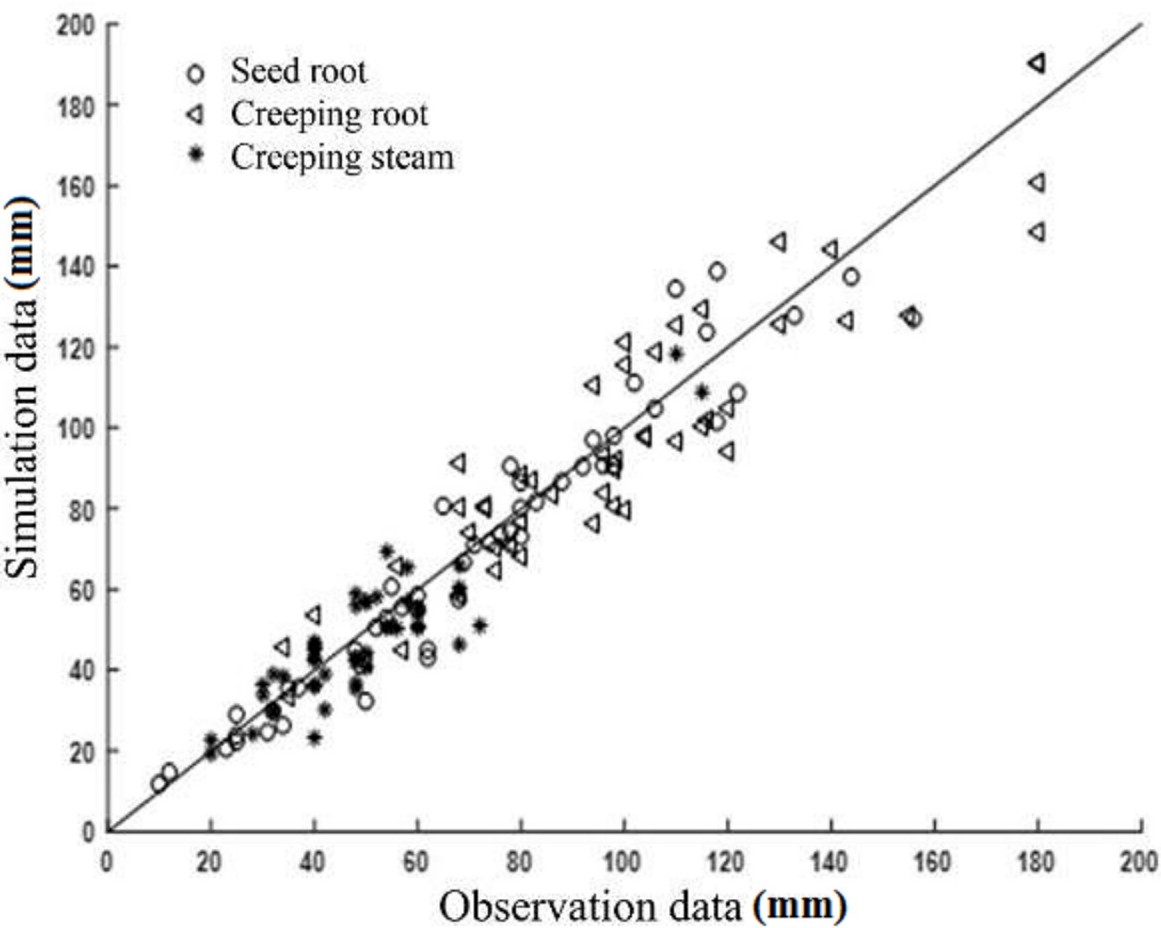

**Fig 18. Comparison of measured and simulated values of Zaodabai.**

and Zaodabai. It can be observed from Table 3 that the RRMSE values of the three root systems of the three varieties were generally distributed between 6.81% and 15.32%, which indicates that the simulated values were consistent with the measured values, and that the simulation results were acceptable. To analyze the cause of deviations, the growth structure of the potato tuber-root system in soil is complicated, which made the measurement difficult and resulted in certain discrepancies. In addition, the bending of the root system is regarded as a uniform change because of the complexity and diversity of the root geometry. Overall, the tuber-root model of the potato constructed by this research scheme is reliable and accurate. This research can form the basis for future studies evaluating the behavior of different root systems. It is recommended that future studies include accurately described root morphology.

## Conclusion

In this paper, a 3D model of the tuber-root system of potatoes based on physical properties was developed. The characterization parameters of the potato tuber-root model at harvesting period were determined. Using three early-maturing potato varieties widely planted in Northeast China as examples, field measurements were performed for the characterization parameters, and a model parameter database was established based on statistical analysis. A 3D visual model of the tuber-root system was constructed and verified by experiments. The results of the

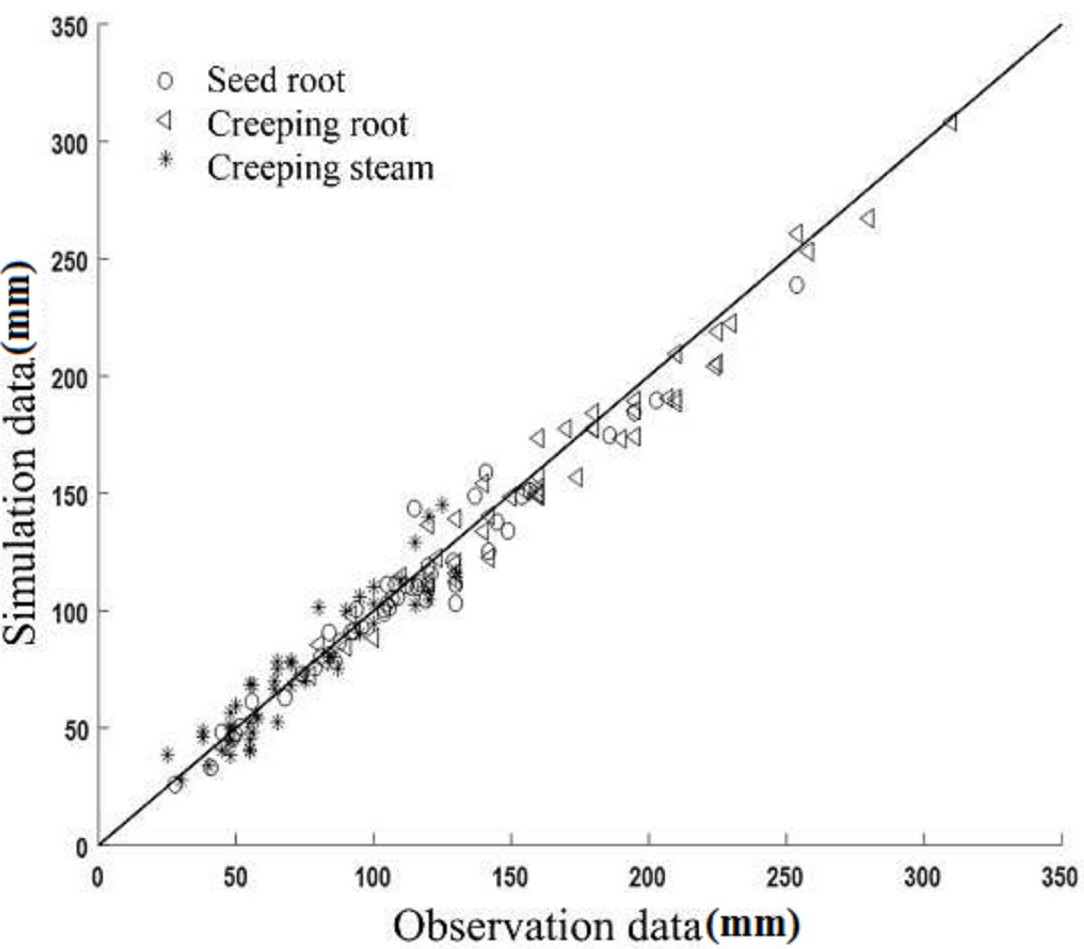

**Fig 19. Comparison of measured and simulated values of Helanshiwu.**

comparative analysis and RRMSE between the measured and simulated values confirmed that the model demonstrated high accuracy and reliability.

Compared with existing modeling methods, the method presented in this paper has the following advantages:

1. The geometric structure description and parameter measurement of the root system were simulated based on the natural growth state of the soil. Moreover, the parameters were determined based on statistical theory, and it was observed that the model was closer to the actual situation and could be applied to the simulation analysis of the interaction between the harvester and soil-tuber-root aggregates.

2. Designing the topological structure based on a standard computer data structure algorithm was conducive to later algorithm development.

3. The simulation method can be applied to the roots of other tuber and rhizome plants such as peanuts and sweet potatoes. It isn't limited to potatoes. It has the characteristics of strong versatility.

To address meet the demands of potato production, the representative parameter databases of the common potato varieties will be established in the future. The method described

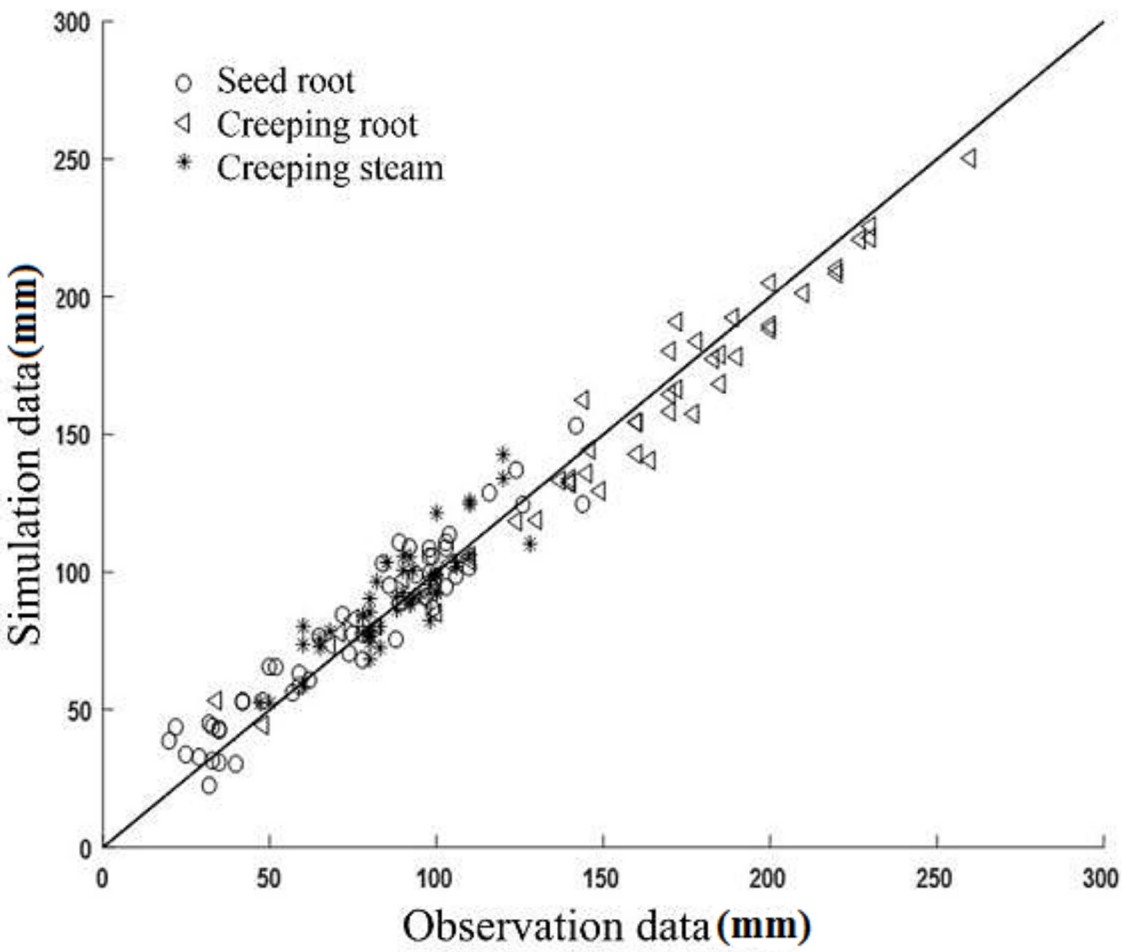

**Fig 20. Comparison of measured and simulated values of Fujin.**

**Table 3. RRMSE results.**

| Variety | RRMSE (%) | | |
|---|---|---|---|
| | Seminal root | Creeping root | Creeping stem |
| Zaodabai | 13.14 | 15.32 | 13.88 |
| Helanshiwu | 8.78 | 6.81 | 13.92 |
| Fujin | 14.23 | 6.84 | 11.37 |

neglected the fiber root because it aimed only at the analysis of the interaction between the fiber and machine tool. Thus, to simulate the actual growth model of the potato more accurately, the next step would be to establish the characteristic parameters of the fiber root in the model.

## Supporting information

**S1 Table. Parameters table of tuber-root model for three varieties of potatoes.** (DOCX)

## Author Contributions

**Conceptualization:** Ping Zhao, Subo Tian.

**Data curation:** Ping Zhao, Yue Tian.

**Formal analysis:** Ping Zhao.

**Funding acquisition:** Ping Zhao, Subo Tian.

**Methodology:** Yue Tian, Guofa Xu, Zichen Huang.

**Project administration:** Ping Zhao.

**Resources:** Ping Zhao, Subo Tian.

**Software:** Ping Zhao, Yongkui Li, Guofa Xu.

**Supervision:** Subo Tian.

**Validation:** Yue Tian, Yongkui Li, Guofa Xu.

**Visualization:** Ping Zhao, Yue Tian.

**Writing – original draft:** Ping Zhao, Subo Tian, Zichen Huang.

**Writing – review & editing:** Ping Zhao, Subo Tian, Zichen Huang.

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
