## [Decision Letter · Decision Letter 0]

17 Jul 2020

PONE-D-20-16069

Potato (Solanum tuberosum L.) Tuber-Root Modeling method Based on Physical Property

PLOS ONE

Dear Dr. Subo Tian,

Thank you for submitting your manuscript to PLOS ONE. After careful consideration, we feel that it has merit but does not fully meet PLOS ONE’s publication criteria as it currently stands. Therefore, we invite you to submit a revised version of the manuscript that addresses the points raised during the review process.

We look forward to receiving your revised manuscript.

Kind regards,

Moumita Gangopadhyay

Academic Editor

PLOS ONE

Additional Editor Comments:

Dear Professor Subo Tian,

Reviewers have now commented on your paper. The paper contains interesting predictive model on potato tuber-root system and very important field of research and here authors have written the paper in an intelligent fashion. However, after reviewing the reviewer’s comments, I make this decision that this paper is not accepted in current form. If the authors are prepared to undertaken the revision required to satisfy all of these following comments, I would be pleased to reconsider my decision:

1. This manuscript needs extensively checking for English language and grammar improvement. It needs your careful reading to be certain that the meaning remained unchanged.

2. Double check all citations against the references list and vice versa with particular attention to matching publication year.

3. In section “3.3.2. Data processing and analysis”, although the method was described clearly, only the result of distributed probability P of initial axial angle APz in the range were given in this paper, others are established with characterization parameter databases. Please do the needful and format your paper according to the template and policy of PLOS.

'The authors gratefully acknowledge the financial support provided by the National Natural Science Foundation of China (NSFC)（51505305）.'

a. Please remove any funding-related text from the manuscript and let us know how you would like to update your Funding Statement. Currently, your Funding Statement reads as follows: 'No'

Reviewers' comments:

Reviewer's Responses to Questions

**Comments to the Author**

1. Is the manuscript technically sound, and do the data support the conclusions?

Reviewer #1: Yes

Reviewer #2: Yes

Reviewer #3: Yes

2. Has the statistical analysis been performed appropriately and rigorously? 

Reviewer #1: Yes

Reviewer #2: Yes

Reviewer #3: Yes

3. Have the authors made all data underlying the findings in their manuscript fully available?

Reviewer #1: Yes

Reviewer #2: Yes

Reviewer #3: Yes

4. Is the manuscript presented in an intelligible fashion and written in standard English?

Reviewer #1: Yes

Reviewer #2: Yes

Reviewer #3: Yes

5. Review Comments to the Author

Reviewer #1: The paper contains interesting predictive model on potato tuber-root system.

The quality of the research work presented in the paper is very good.

The paper is scientifically sound.

However, In conclusion part explanation of the result obtain in model validation will be much appreciated.

Reviewer #2: Tuber root modelling studies are very important field of research and here authors have written the paper in an intelligent fashion. also the data furnished are sufficient.Hence this paper can be accepted.

Reviewer #3: This paper mainly studied a method of 3D model construction of the potato tuber root system based on physical properties, which is important and difficult. So, this research is  relatively innovative. Experiment results show that the proposed method has good performance. The research work is suggestive and the point of view is worthy of researching. Data and analysis are enough to support conclusions. However, for the benefit of reader, some revisions are needed in this paper, there are given below.

First, some sentences contain grammatical and spelling mistakes, such as the title ”........Physical Property”, I think that “Property” should be “Properties”, et al . in addition, the same expression should be consistent, such as “early maturing” in abstract and “early-maturing” in text. Please check and correct carefully at revision, and pay more attention to English grammar and sentence structure.

Second, in section “3.3.2. Data processing and analysis” , although the method was described clearly, only the result of distributed probability P of initial axial angle APz in the range were given in this paper, others are established with characterization parameter databases. But the characterization parameter databases weren’t provide as appendix table, Add or not, please according to the editor's request. I think the data are full and perfect, because you(the authors ) said “The data used to support the findings of this study are available from the corresponding author upon request” when you submit the manuscript.

Finally, There are errors in writing, such as , in row 235th and 236th, A (x1, y1, z1) should be A(x1, y1, z1), B (x1, y1 + l, z1) should be B(x1, y1+l, z1), et al. Please check and correct carefully at revision.

Format your paper according to the template and policy of PLOS.

6. PLOS authors have the option to publish the peer review history of their article (what does this mean?). If published, this will include your full peer review and any attached files.

Reviewer #1: **Yes: **Arpita Das

Reviewer #2: No

Reviewer #3: No

---

## [Author Response · Author response to Decision Letter 0]

14 Aug 2020

Dear editor and reviewers: 

Thank you for your comments concerning our manuscript entitled “Potato (Solanum tuberosum L.) tuber-root modeling method based on physical properties”. (ID: PONE-D-20-16069). Those comments are all valuable and extremely helpful for revising and improving our paper. We studied the comments carefully, and made corrections and improvement which we hope meet with approval. The main changes are in the manuscript and the responses to the reviewers’ comments are as follows. 

Additional Editor Comments:

1. This manuscript needs extensively checking for English language and grammar improvement. It needs your careful reading to be certain that the meaning remained unchanged.

Response1:We extensively checking for English language and grammar improvement with the help of Editage (www.editage.cn), and meaning remained unchanged. 

2.Double check all citations against the references list and vice versa with particular attention to matching publication year.

Response2:We modified the reference using Zotero, and checked citations with the format on webpage (https://journals.plos.org/plosone/s/submission-guidelines).

3. In section “3.3.2. Data processing and analysis”, although the method was described clearly, only the result of distributed probability P of initial axial angle APz in the range were given in this paper, others are established with characterization parameter databases. Please do the needful and format your paper according to the template and policy of PLOS.

Response3:We provided the other characterization parameters in Supporting Information file. 

1.Please ensure that your manuscript meets PLOS ONE's style requirements, including those for file naming. 

Response1:We revised the manuscript’s style according to the PLOS ONE style templates.

2.We note that you have indicated that data from this study are available upon request. PLOS only allows data to be available upon request if there are legal or ethical restrictions on sharing data publicly. For more information on unacceptable data access restrictions, please see http://journals.plos.org/plosone/s/data-availability#loc-unacceptable-data-access-restrictions.

Response2:We are very sorry for our wrong input (“No-some restirictions will apply”) when we submitted the manuscript. The reason is that we didn’t understanding the explanations well.

We should input as follow: 

Yes - all data are fully available without restriction. 

As well as we provide the supporting data according to the requirement of the “Additional Editor Comments” as Supporting information file. Please you decide Whether the data is published in the manuscript as an appendix.

3.Thank you for stating the following in the Acknowledgments Section of your manuscript:

'The authors gratefully acknowledge the financial support provided by the National Natural Science Foundation of China (NSFC)（51505305）.'

 a. Please remove any funding-related text from the manuscript and let us know how you would like to update your Funding Statement. Currently, your Funding Statement reads as follows: 'No'

Response3:We removed the funding-related text from the manuscript.

We would like to update our funding statement on the webpage. Our Funding Statement should be “Yes”. Please help us to change the online submission as follow. 

4.Financial Disclosure. Enter a financial disclosure statement that describes the sources of funding for the work included in this submission. Review the submission guidelines for detailed requirements. View published research articles from PLOS ONE for specific examples. 

This statement is required for submission and will appear in the published article if the submission is accepted. Please make sure it is accurate.

Response4:The present work is financially supported the National Natural Science Foundationof China(NSFC) of Zhao Ping, and the grant mumber is 51505305. Zhao Ping had an important role in the study design, data collection and analysis, decision to publish, and preparation of the manuscript.

Reviewer #1: 

1.The paper contains interesting predictive model on potato tuber-root system.The quality of the research work presented in the paper is very good.

The paper is scientifically sound.

Response1:We appreciate the reviewer’s view of the article.

2.However, In conclusion part explanation of the result obtain in model validation will be much appreciated.

Response2:We supplemented some explanation of the result in section “Conclusion”, and marked in red in the file named “Revised Manuscript with Track Changes”.

Reviewer #2: 

1.Tuber root modelling studies are very important field of research and here authors have written the paper in an intelligent fashion. also the data furnished are sufficient. Hence this paper can be accepted.

Response1:We appreciate the reviewer’s view of the article.

Reviewer #3: 

1.This paper mainly studied a method of 3D model construction of the potato tuber root system based on physical properties, which is important and difficult. So, this research is relatively innovative. Experiment results show that the proposed method has good performance. The research work is suggestive and the point of view is worthy of researching. Data and analysis are enough to support conclusions. However, for the benefit of reader, some revisions are needed in this paper, there are given below.

Response1:We appreciate the reviewer’s view of the article.

2.First, some sentences contain grammatical and spelling mistakes, such as the title ”........Physical Property”, I think that “Property” should be “Properties”, et al . in addition, the same expression should be consistent, such as “early maturing” in abstract and “early-maturing” in text. Please check and correct carefully at revision, and pay more attention to English grammar and sentence structure.

Response2:We replaced “Property” with “Properties”,and replaced “early-maturing” with “early maturing”. in addition, we extensively checked English language and grammar and revised them.

3.Second, in section “3.3.2. Data processing and analysis” , although the method was described clearly, only the result of distributed probability P of initial axial angle APz in the range were given in this paper, others are established with characterization parameter databases. But the characterization parameter databases weren’t provide as appendix table, Add or not, please according to the editor's request. I think the data are full and perfect, because you(the authors ) said “The data used to support the findings of this study are available from the corresponding author upon request” when you submit the manuscript.

Response3:We provided all the characterization parameter data as a supporting information file. We obey the editor's arrangement, data are fully available without any restriction, and we would like to make the data public. 

4.Finally, There are errors in writing, such as , in row 235th and 236th, A (x1, y1, z1) should be A(x1, y1, z1), B (x1, y1 + l, z1) should be B(x1, y1+l, z1), et al. Please check and correct carefully at revision.

Response4:We modified as “Supposing that the starting node of a root is A(x1, y1, z1), and the original growth trajectory is to grow a unit length l along the Y axis to the point of B(x1, y1 + l, z1)…” and checked carefully all the text and corrected the errors in writing.

5.Format your paper according to the template and policy of PLOS.

Response5:We already format our paper according to the given template and policy of PLOS. 

 We tried our best to improve the manuscript and made some changes in the manuscript. These changes will not influence the content and framework of the paper. And here we did not list the changes but marked in red in revised paper.

We appreciate for Editors/Reviewers’ warm work earnestly, and hope that the correction will meet with approval.

Once again, thank you very much for your comments and suggestions.

Thank you and best regards.

Yours sincerely, Professor Subo Tian

tiansubo@syau.edu.cn

College of Engineering, Shenyang Agricultural University

---

## [Editor Report · Decision Letter 1]

31 Aug 2020

Potato (Solanum tuberosum L.) tuber-root modeling method based on physical properties

PONE-D-20-16069R1

Dear Dr. Tian,

We’re pleased to inform you that your manuscript has been judged scientifically suitable for publication and will be formally accepted for publication once it meets all outstanding technical requirements.

Kind regards,

Moumita Gangopadhyay

Academic Editor

PLOS ONE

Additional Editor Comments (optional):

Thank you very much for answering all the comments made by the reviewers. The necessary modifications of the manuscript now seem appropriate in terms of scientific means and also English language corrections. Therefore, I am happy to inform you that this paper can be accepted in the current form for publications. Thank you again for choosing this journal for your valuable manuscripts.
---

## [Editor Report · Acceptance letter]

8 Sep 2020

PONE-D-20-16069R1 

Potato (*Solanum tuberosum* L.) tuber-root modeling method based on physical properties 

Dear Dr. Tian:

I'm pleased to inform you that your manuscript has been deemed suitable for publication in PLOS ONE. Congratulations! Your manuscript is now with our production department. 

Kind regards, 

on behalf of

Dr. Moumita Gangopadhyay 

Academic Editor

PLOS ONE